# Course of Adverse Events during Short Treatment Regimen in Patients with Rifampicin-Resistant Tuberculosis in Burundi

**DOI:** 10.3390/jcm9061873

**Published:** 2020-06-16

**Authors:** François Ciza, Tinne Gils, Michel Sawadogo, Tom Decroo, Alberto Roggi, Alberto Piubello, Nimer Ortuño-Gutiérrez

**Affiliations:** 1Damien Foundation, Burundi 15, Avenue du Poisson 15, Ntahangwa, Bujumbura 2426, Burundi; frciza@gmail.com (F.C.); sawadgom@gmail.com (M.S.); 2Department of Clinical Sciences, Institute of Tropical Medicine, Nationalestraat 155, 2000 Antwerp, Belgium; tdecroo@itg.be; 3Research Foundation Flanders, Egmonstraat 5, 1000 Brussels, Belgium; 4Damien Foundation, Boulevard Léopold-II 263, 1081 Brussels, Belgium; alberto.roggi@damien-foundation.be (A.R.); nimer.ortunogutierrez@damiaanactie.be (N.O.-G.); 5Damien Foundation, POBox 1065, Niamey, Niger; albertopiubello@yahoo.it

**Keywords:** multi-drug-resistant tuberculosis, short treatment regimen, Burundi

## Abstract

The introduction of the nine-month short-treatment regimen (STR) has drastically improved outcomes of rifampicin-resistant tuberculosis (RR-TB) treatment. Adverse events (AE) commonly occur, including injectable-induced hearing loss. In Burundi we retrospectively assessed the frequency of adverse events and treatment modifications in all patients who initiated the STR between 2013–2017. Among 225 included patients, 93% were successfully treated without relapse, 5% died, 1% was lost-to-follow-up, 0.4% had treatment failure and 0.4% relapsed after completion. AE were reported in 53%, with grade 3 or 4 AE in 4% of patients. AE occurred after a median of two months. Hepatotoxicity (31%), gastro-intestinal toxicity (22%) and ototoxicity (10%) were most commonly reported. One patient suffered severe hearing loss. Following AE, 7% of patients had a dose reduction and 1% a drug interruption. Kanamycin-induced ototoxicity led to 94% of modifications. All 18 patients with a modified regimen were cured relapse-free. In this exhaustive national RR-TB cohort, RR-TB was treated successfully with the STR. Adverse events were infrequent. To replace the present STR, all-oral regimens should be at least as effective and also less toxic. During and after transition, monitoring, management, and documentation of AE will remain essential.

## 1. Introduction

Tuberculosis (TB) is among the lead causes of deaths caused by antimicrobial resistance. While 484,000 people developed drug rifampicin-resistant tuberculosis (RR-TB) in 2018, only 187,000 were detected and notified, and 156,000 started treatment. Globally, RR-TB treatment success was 56% [1]. Short treatment regimens (STR), based on the so-called “Bangladesh regimen”, showed over 80% treatment success [2,3], compared to 75% of patients treated with longer individualised regimens [4]. 

Occurrence of adverse events (AE) deemed “severe” varies between 5.6% and 48.2% in patients on STR [5,6]. Among drugs used in STR, second-line injectables have the highest risk of severe AE leading to treatment discontinuation (capreomycin (Cm) 8.2%, kanamycin (Km) 7.5%), while others have a lower risk (moxifloxacin (Mfx) 2.9%, clofazimine (Cfz) 1.6%, levofloxacin 1.3%) [7]. Irreversible hearing loss resulting from injectable use is one of the most debilitating events [7,8]. In a rapid communication, the WHO therefore recommended phasing out injectable drugs in favor of all-oral short or long individualised regimens containing bedaquiline, depending on baseline resistance [9]. The long-term safety and effectiveness of these regimens, and their potential to prevent acquired resistance to bedaquiline, is still to be established [10]. Meanwhile, appropriate AE management in STR is essential.

Burundi started RR-TB treatment with a nine-month STR in May 2013, as part of a nine-country African study, with 91.8% treatment success. In this study, 89.2% of patients had an AE, 10.7% a severe AE, and 7.1% ototoxicity [11]. Several studies have looked at incidence and grading of AE in RR-TB treatment [7,12]. However, little evidence exists on AE grading over time, frequency, and timing of regimen modification in terms of dose reductions or treatment interruptions, and subsequent changes of AE grading. We aimed to describe the time course of AE, regimen modifications and resulting changes in AE grading in patients who started the RR-TB STR between May 2013 and December 2017 in Burundi. We also describe characteristics and treatment outcomes for these patients.

## 2. Materials and Methods 

This is a retrospective analysis of routine program data of RR-TB patients in Burundi. All patients who initiated STR between May 2013 and December 2017 in Burundi were included. Data were collected until October 2019. Eligibility for STR was RR-resistance by Xpert^®^ Mycobacterium Tuberculosis/Rifampicin (Xpert MTB/RIF) (Xpert; Cepheid, Sunnyvale, CA, USA), drug susceptibility testing (DST), and no previous treatment with second-line drugs, nor known fluoroquinolone (FQ) or injectable resistance. Additionally, written voluntary consent was required for the prospective study between May 2013–May 2015, including for hospitalization for the study duration. During that period, additional exclusion criteria were age <15 years, pregnancy, a pre-treatment electrocardiogram (ECG) with a QT-interval of >500 ms, or exclusion based on assessment by the principal investigator [11]. 

Burundi is a land-locked country in Central Africa, with a population estimated at 11 million in 2018 [13]. In Burundi, the respective incidences of TB and RR-TB are about 111 and 3.3 per 100000 population and 2600 patients are estimated to have died from TB in 2018 [1]. All TB-patients are followed by the National TB-programme of Burundi. There are 308 TB-treatment centers, 170 of which provide smear microscopy. The national reference laboratory performed culture and DST. RR-TB patients are referred to the national reference center for RR/multi-drug-resistant (MDR)-TB of Kibumbu (Centre National de Reference des Tuberculeux Rifampicin/Multirésistants de Kibumbu; CNPEC-TB/MR). 

Smears and cultures of sputum specimens were collected before treatment initiation, and smears monthly during treatment (Table 1). Culture was performed on Löwenstein-Jensen solid medium. Follow-up after cure continued at least up to one year after treatment completion with smear and culture performed every six months. Xpert MTB/RIF testing became available in 2012 at the national reference laboratory and was performed on TB-patients with a history of TB-treatment or known RR-TB contacts. During the prospective study, samples were also sent to the supra-national reference laboratory of the Institute of Tropical Medicines, Antwerp (ITM) for DST [11]. 

All patients initiated a regimen containing Kanamycin (Km), normal-dose Moxifloxacine (Mfx), Prothionamid (Pto), Clofazimine (Cfz), Pyrazinamid (Z), high-dose Isoniazid (Hh) and Ethambutol (E) in the four-to-six-month intensive phase (duration depending on time of smear conversion), and Mfx-Cfz-Z-E in the five-month continuation phase. 

Patients were examined at baseline and monthly after initiation for bacteriological progress and regularly for occurrence of AE (Table 1). Patients enrolled in the trial were hospitalised at the CNPEC-TB/MR for the treatment duration and re-assessed every six months for two years after completion. All other patients were hospitalised during the intensive phase, received daily treatment in a TB-treatment center in their community during the continuation phase, and were re-assessed at 6 and 12 months after completion. Patients who came for post-treatment follow-up during the trial, and for daily ambulatory treatment afterwards, received a compensation for transport costs. 

AEs, detected during routine examinations (Table 1), were recorded monthly and graded using the scale of the French National Agency of Research on AIDS (Agence Nationale de Recherche sur le SIDA; ANRS) [14]. Hearing loss was measured in decibels (dB) using pure tone audiometry. The weighted average hearing loss was calculated and classified as the following grades of hearing loss compared to baseline: 0 (normal), 20 dB; 1 (mild): 21–40 dB; 2 (moderate): 41–70 dB; 3 (severe): 71–90 dB; 4 (very severe) >90 dB. Hearing loss of any grade was considered irreversible. Ototoxicity was managed by reducing the frequency of Km from daily to thrice a week and by replacing Km with capreomycin (Cm) in case of no improvement. 

We defined cured as smear converted, and time until smear conversion as the number of days until the earliest of two consecutive negative smear results of at least 30 days apart since treatment initiation. Other outcomes were defined per WHO guidelines [15]. Relapse-free success included cure or treatment completion without evidence or relapse one year after treatment completion. Programmatically adverse outcomes included lost-to follow-up (LTFU), death, treatment failure and relapse. Clinically adverse outcomes included death, treatment failure and relapse. Bacteriologically adverse outcomes included treatment failure and relapse. AE were defined and graded per national protocol, based on the ANRS scale: Grades range 1–4, representing 1: mild, 2: moderate; 3: severe and 4: life-threatening or permanently disabling events [14]. Body mass index (BMI) groups were defined as follows: <16.0 kg/m2 severely underweight, 16.0–18.4 underweight, 18.5–24.9 normal, 25.0–29.9 overweight, ≥30 obese. 

Programme data were collected by trained care providers on individual paper-based RR-TB registers stored at CNPEC-TB/MR. A copy followed the patient to ambulatory sites after the intensive phase, and updates were returned to the CNPEC-TB/MR, where they were entered into EpiData Entry database (version 3.1; EpiData Association, Odense, Denmark). Ambulatory care providers were trained in data collection, and the national RR-TB focal point conducted regular supervisory visits to ambulatory sites. A study database limited to study variables was constructed for analysis and de-identified data were shared with the analyst at ITM. Inconsistencies were solved through communication with the study team in Burundi and verification of the source documents. 

Stata software (version 16.0; StataCorp LP, College Station, TX, USA) was used for analysis. Proportions were used to summarize categorical variables and medians, interquartile ranges (IQR), and ranges were used for continuous variables. 

The study was approved by the National Ethics Committee of Burundi and the Institutional Review Board of the Institute of Tropical Medicines (ITM) in Antwerp, Belgium.

## 3. Results

### 3.1. Characteristics at Initiation

237 TB-patients were identified as RR between May 2013 and December 2017. Twelve (5.1%) patients were excluded; five (2.1%) had been exposed to second-line drugs, one (0.4%) was LTFU before start, and one (0.4%) was confirmed drug-susceptible after DST. Three (1.3%) minors (<15 years old) and two (0.8%) patients who did not consent (one was comatose, one refused hospitalization) were excluded during the prospective study. 225 (95.3%) among RR-TB patients initiated the STR in Burundi between May 2013 and December 2017 (Table 2 and Table 3). 217 (96.4%) were included based on a positive Xpert MTB/RIF, and eight (3.6%) based on phenotypic DST-results only. Among 60 (26.7%) patients with a baseline DST result available, DST showed RR in all (100%), resistance to H in 56/59 (94.9%), resistance to streptomycin in 14/24 (58%), resistance to E in 12/18 (66.7%), susceptibility to Km in 38/38 (100%), and susceptibility to FQ in 37/37 (100%). All patients had pulmonary TB and none had taken second-line TB-medication before. Among patients with an X-ray result, 59 (30.4%) showed one affected lobe only, 80 (41.2%) two or three lobes affected, and 55 (28.4%) four or more lobes affected. Five HIV-positive patients who were not on anti-retroviral treatment (ART) at presentation started ART after a median of 16 (IQR: 14–23) days on STR. Culture results at initiation were available for 111 (49.3%) of patients, among who 83 (74.8%) were positive, 15 (13.5%) negative, and 13 (11.7%) scanty or contaminated.

### 3.2. Outcomes

Of 225 patients, 209 (92.9%) were cured (185, 82.2%) or completed treatment (25, 10.7%) without evidence of relapse, one (0.4%) experienced treatment failure, three (1.3%) were lost-to-follow-up (LTFU), 11 (4.9%) died during treatment and one (0.4%) relapsed six months after being declared cured (Table 4). 

Among those with relapse-free success, duration of treatment was nine (median, IQR: 9–9) months. Cured patients smear converted after one (median, IQR: 1–2) month. Culture results were incomplete at baseline and culture was not done monthly, but all available cultures for those with relapse-free cure were negative at six (*n* = 202) and nine (*n* = 190) months. The patient with treatment failure was HIV-positive and severely underweight. He smear converted after six months, but reconverted at eight months. Three patients who were LTFU had smear converted after one (median, IQR: 1–1) month. Two had mild hepatotoxicity, which resolved at month three and five. Among 11 patients who died, five (45.5%) died within two months after initiation. Four (36.4%) among the dead were HIV positive, and one was initiated on ART after enrolment. Four (36.4%) patients had a grade 1 AE before death, and three of which were also on ART (one neurological, one gastro-intestinal, one ototoxic event). Among 210 patients completing treatment, 198 (94.7%) had at least one follow-up visit and 81 (38.6%), 176 (83.8%), 53 (25.2%) and 43 (20.5%) had a follow-up visit 6, 12, 18 and 24 months after completion, respectively. Among patients returning after one year, all who had a smear (176, 100.0%) or culture (59, 33.5%) had a negative result. Two patients died seven days and six months after completion, the first one of pharyngeal cancer. We found no significant associations between having any AE, having any severe AE, or HIV-status with adverse outcomes. 

### 3.3. Occurrence of Adverse Events

Overall, 119 (52.9%) patients had at least one AE and nine (4.0%) had a grade 3 or 4 event (Table 5). The most commonly occurring events were hepatotoxicity (30.7%), gastro-intestinal toxicity (21.8%), and ototoxicity (10.2%). Only hepatotoxicity led to two grade 4 events (0.9%) after two months. Grade 3 events were hepatotoxic (4, 1.8%), ototoxic (1, 0.4%), dermato-toxic (1, 0.4%) or gastro-intestinal (1, 0.4%). Nephrotoxic, neurotoxic and osteo-arthritic events and headache (five patients (2.2%), after four (median, IQR: 2–5) months) only occurred as mild events (Figure 1 and Figure 2).

### 3.4. Course of Adverse Events and Regimen Modifications

Among all 225 patients, 24 (10.6%) had a regimen change (18, 8.0%) and/or a grade 3 or 4 AE (9, 4.0%) (Figure 3). Among treatment modifications, two (11.1%, grade three) followed severe AE and 16 (88.8%) grade 1 (14, 66.7%) or 2 (2, 11.1%) events. Modifications followed only ototoxicity (17, 94.4%) or dermato-toxicity (one, 5.6%) and happened two (median, IQR: 1–3) months after treatment initiation. Seven (77.8%) grade 3 or 4 AE did not lead to treatment modifications. Permanent discontinuation of any drug happened in three (1.3%) patients after one (median, IQR: 1–4) month on treatment. Dose reductions happened in 16 (7.1%) patients after two (median, IQR: 2–3) months on treatment. All eight HIV-positive patients with a grade 3 or 4 AE and/or regimen modifications were on ART at enrolment, including two patients with severe hepatotoxicity.

Of 175 reported AE, 155 (88.6%), 12 (7.4%)5 (2.9%), and two (1.1%) first presented as grade 1, 2, 3 or 4, respectively. Eight (4.6%) AE increased in severity after presentation, all of which initially presented as grade 1, while 167 (95.4%) AE decreased in severity. The maximum grading after increase was grade 2 (6, 75.0%) or 3 (2, 25.0%). 

Among 69 hepatotoxic events, 53 (76.8%), 10 (14.5%), 4 (5.8%) and 2 (2.9%) presented as grade 1, 2, 3 or 4, respectively. 4/53 (7.5%) evolved from grade 1 to 2 after two (median, IQR: 2–2) months, remaining at grade 2 for one (median, IQR: 1–1) month before reducing. For hepatotoxic events of grade 2, 3 and 4, grading decreased after one (median, IQR: 1–2) month. 

21/23 (91.3%) ototoxic events presented as grade 1, and one evolved to irreversible hearing loss of 71–90 dB after a month. The patient was HIV-positive, had baseline ototoxicity, and a history of category two treatment failure. The frequency of Km was reduced at month one, and Km was stopped in month three. Overall, 17/23 (73.9%) ototoxic events led to modified administration of Km; 16 (94.1%) reductions in frequency of administration after two (median, IQR: 2–3) months; and two (11.8%) permanent discontinuations after one and four months, in which Cm replaced Km. Two (8.7%) patients presented with grade 2 hearing loss. Six (26.1%) patients with ototoxicity had previously been treated with streptomycin, and eight (34.8%) were HIV-positive. Among patients with treatment modifications, five (29.4%) had streptomycin exposure, three (17.6%) had baseline ototoxicity and six (35.3%) were HIV-positive. Clinicians did not change treatment after six (26.1%) self-limiting reports of tinnitus (grade 1).

Ten (of 11, 90.9%) dermatoxic events presented as grade 1 and resolved after one (median, IQR: 1–3) month. One (9.1%) severely underweight patient with throat cancer presented with grade 3 generalized skin eruptions after two months, at which Hh and Pto were discontinued and the AE resolved. The patient continued the same standard regimen without Hh and Pto, was declared cured of TB after nine months, but died one week later. There were no other treatment modifications due to dermato-toxicity. 

48/49 (98.0%) patients with gastro-intestinal toxicity presented with grade 1, two of which evolved to grade 2 after one month and resolved one to two months later, and one evolved to grade 3 for one month before resolving. One (2.0%) patient had grade 2 gastro-intestinal toxicity at detection, which resolved after one month. Gastro-intestinal toxicity did not trigger regimen modifications. Mild nephrotoxicity, neurotoxicity, osteo-articular toxicity and headache resolved after one (median, IQR: 1–2) month, without treatment modifications. 

## 4. Discussion

Our study included all RR-patients who initiated STR in Burundi over a 4.5-year period and describes in detail the course and management of AE. More than nine out of ten eligible patients in Burundi were enrolled and had excellent treatment outcomes; 92.9% had relapse-free cure. Severe AE were uncommonly reported, occurred early and all but one resolved quickly. Only 4.5% of AE increased in grade after onset, and all others decreased. Treatment adaptations were infrequent and mainly followed ototoxicity.

Programmatic STR-outcomes in this challenging setting compare favorably with pooled estimated of STR (80.0%), and long individualised RR-regimens (75.3%) [2,4]. LTFU (1.3%) was low and happened late in the continuation phase, after patients had smear converted. One patient left the country. The other two patients, with a recovered health status, may have been reluctant to continue showing up for the required daily observed treatment [16]. The death rate (4.9%) was relatively low. Deaths mostly happened early, indicating advanced TB, and some were likely attributable to co-morbidities, such as advanced HIV and cancer [17]. Rates of patients with bacteriological failure (0.4%) or relapse (0.4%) were low [2].

In our Mfx-based STR cohort, outcomes approach those from gatifloxacin (Gfx)-based cohorts. Gfx is more effective compared to other FQ in STR-regimens; successful outcomes ranged between 85–96% in cohorts on Gfx-based STR, compared to 80–81% in cohorts on Mfx-based STR [2,18]. The patients from Burundi included in the African nine-country study also performed better compared to other countries [11]. We speculate that this may be explained by a lower frequency of initial FQ-resistance, known to be the strongest predictor of unfavorable outcomes in African studies [2,4]. We do not have baseline resistance data and can thus not estimate the true burden of baseline FQ-resistance. Variations of the Bangladesh regimen have been used in Burundi since 2002, initially a ofloxacin-based 15-month regimen, and since 2012 a Gfx-based 12-month regimen, with over 90% treatment success [19]. The use of standardized and effective treatment regimens, with a superior FQ, has likely limited development of resistance and limited subsequent transmission in the community [18,20]. Four (36.4%) among 11 patients who died had HIV, and the lack of an association between HIV-status and having an adverse outcome contrast with findings from other studies [2,11,21]. This may be explained by the low number of HIV coinfected patients in our cohort. In our cohort, overall occurrence of any reported AE (53.1%) and severe or debilitating AE (4.0%) were low. The nine-country study reported any AE in 89.2% and grade 3 or 4 AE in 10.7% of patients on STR, respectively [2,11]. Piubello et al. reported 78.7% with any AE, and 6.6% with either a grade 3 or 4 AE (3.2% grade 3 and 2.4% grade 4), respectively, in Niger [6]. The Standard Treatment Regimen of Anti-tuberculosis Drugs for Patients With MDR-TB (STREAM)prospective trial reported 48.2% of patients with grade 3 or higher in STR, including cardiac disorders in 10.6% of patients [5]. Review of cohorts with >20% HIV-positive patients on RR-treatment showed 83% and 24% of patients who experienced any event, or a grade 3 or 4 event, respectively. Consistent with these studies, we found no increased risk of AE in HIV-positive patients [12]. Differences may be explained by differences in regimens, AE definitions, monitoring, and reporting in routine versus trial settings. Our data approach those of Niger, where similar regimens and equal definitions were used [6].

In Burundi, hepatotoxic events were the most reported causes of having an AE and having a severe AE, yet mild forms were less commonly reported than in other studies [6,11]. Gastro-intestinal events were more common in other studies (54.6–57.1%) compared to Burundi, where light meals prior to drug administration during the intensive phase may have prevented disturbances [6,11,12]. The most debilitating AE were hearing loss and tinnitus, occurring in one in ten patients, less frequent than in other studies on STR (25.3–44.3% [3,4]) and 36% in other cohorts with >20% HIV-positive patients [12]. Regular audiograms were taken at the same frequency compared to the nine-country study (at initiation and month four), but less frequent than in Niger (also at month two) [6,11]. The low numbers could partly be explained by the lower proportion of patients with previous streptomycin exposure in our study. Clinicians reacted quickly to ototoxicity, changing treatment promptly after AE presentation and reducing dosage or interrupting Km in 7.6% of patients, very similar to what was shown by a recent meta-analysis from 28 countries. The latter showed 7.5% Km-discontinuation in patients using this drug, mainly following ototoxicity [7]. Despite the response, one patient suffered severe irreversible hearing disorders. Ototoxicity from aminoglycosides is dose-dependent, but it is unclear whether reduction of the cumulative weekly dose when hearing loss is present prevents its progression [8,22]. More effective than dose reduction is prompt replacement of Km with other drugs. In Niger, the combination of monthly audiometry with replacement of Km by linezolid was effective in preventing severe hearing loss [6]. Another study shows that bedaquiline can also be used to replace Km [23]. However, bedaquiline resistance is acquired in up to 5% of patients [24,25], which is used as an argument to save this drug for patients with FQ-resistant-TB [26]. All 17 patients with a Km dose reduction or withdrawal of Km from the regimen, which occurred after a median of two months on treatment, were cured relapse-free. The patient with severe skin eruptions leading to Pto and H interruption (0.4%) was cured from RR-TB but died from cancer. In the meta-analysis, the risk of discontinuation of Pto (or ethionamide) was 6.5%, mainly following gastro-intestinal or hepatoxic AE [7].

The limitations of our study are mainly linked to its retrospective nature. Many patients had no culture results at baseline, and we used smear conversion as interim outcome. The use of culture was limited due to problems with maintenance of the biosafety-cabinet at the national reference laboratory. Such situations are common in resource-limited settings, showing the need for adapted tools to measure treatment response [27]. However, smear-based results were likely to be reliable as the laboratory network has a well-established External Quality Assurance system. In those declared cured on smears, post-treatment follow-up showed that all except one remained relapse-free. We did not have complete baseline resistance data and could thus not analyze its impact on outcomes or AE. It is likely that baseline resistance to FQ was low, contributing to the excellent outcomes.

Part of the low reported occurrence of AE and low frequency of treatment modifications in Burundi could be explained by under detection, due to the fact that AE monitoring was less stringent compared to prospective trials. In trials more resources are usually available, which allow more frequent and more systematic monitoring, and less reporting bias. In addition, post-mortem investigations were not regularly performed to establish cause of death. Seeing that ECG was not regularly monitored, cardiac toxicity leading to death could have been under detected. Liver enzymes tests were done once after baseline if no other indications, explaining why hepatotoxic events were concentrated around month two. Mild hepatotoxic and other self-resolving events might have been underreported. For 15 patients, mild or moderate hearing loss was no longer reported after lowering the dose of Km. While vestibular toxicity can resolve, reversible hearing loss following treatment with injectable aminoglycosides is not possible [22,28]. We verified patients’ files, which confirmed the findings. We speculate that these results reflect a measurement error. The procedure involves calculation of a weighted average of hearing loss in each ear based on results from audiometry, complemented with clinical examination, and comparison with baseline results. We speculate that clinicians stopped reporting existing hearing loss once it stopped deteriorating compared to baseline. For the purpose of this study we revised the grading considering all hearing loss irreversible.

Our study has important strengths; we analysed an exhaustive national sample, with a high inclusion rate. LTFU was low (1.3%) and late, further reducing selection bias. We had a very high coverage of post-treatment follow-up; 94.7% had at least one follow-up visit, and 83.8% were followed a year or more after treatment ended. This is much higher compared to the African nine-country study, where only 57.2% of patients were followed until a year after completion [3]. Any missing data, inconsistencies or doubts were verified by checking the original patient files. We can thus be relatively confident that our data reflect the reality of routine RR-TB care in Burundi.

The WHO recommends the phasing out of injectable-based STR in favor of all-oral regimens containing bedaquiline for RR-TB [9]. New regimens often combine FQ and bedaquiline, and linezolid. In individual patients’ meta-analysis, the pooled incidence of AE leading to treatment modifications from levofloxacin was 1.3%, bedaquiline 1.7%, Mfx 2.9% and linezolid 14.1%. Bedaquiline and Mfx can both cause cardiotoxicity, linezolid peripheral neuropathy and myelosuppression [7]. In the STREAM trial, 11.3% of patients had a treatment change due to Mfx-induced cardiac toxicity [5]. Without injectables, AE monitoring will remain essential to reduce debilitating morbidity and mortality during RR-treatment. Documentation of adverse event occurrence and management will be necessary to inform policy and practice.

A regimen should not only be tolerable, but also effective in preventing acquired resistance. Whereas resistance acquisition after treatment with injectable-containing regimens has been reported for most shorter treatment regimen studies [2], this feature was not considered in the ranking of second-line TB drugs shown in the 2019 WHO guidelines [9]. The potential of acquiring bedaquiline resistance in regimen with at least four likely active drugs [24,25] shows that prevention of resistance acquisition requires more attention as all-oral regimens are being investigated.

Our study showed that successful treatment of RR-TB with short regimens is feasible in Burundi, and that clinicians adequately managed AE. Awaiting more tolerable and also effective injectable-free regimens, prompt interruption of injectables at occurrence of AE could prevent further hearing loss in patients treated with second-line injectable containing STR.

## Figures and Tables

**Figure 1 jcm-09-01873-f001:**
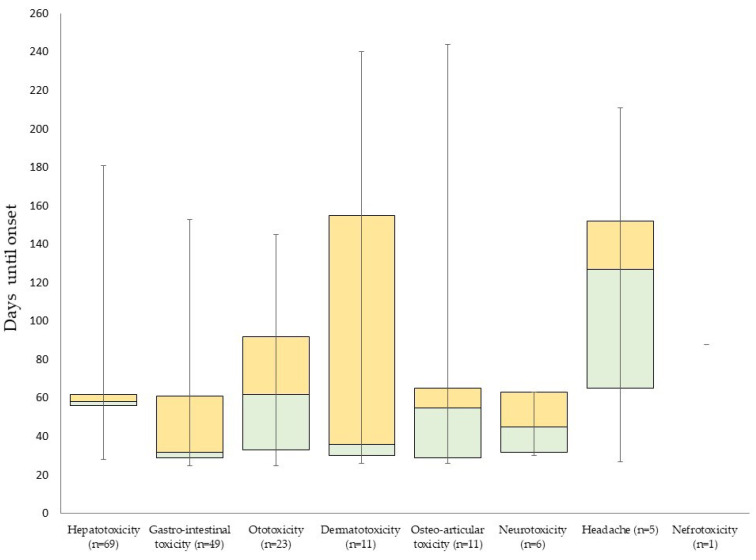
Boxplot of time until onset of adverse events in patients on short regimens for rifampicin-resistant tuberculosis in Burundi.

**Figure 2 jcm-09-01873-f002:**
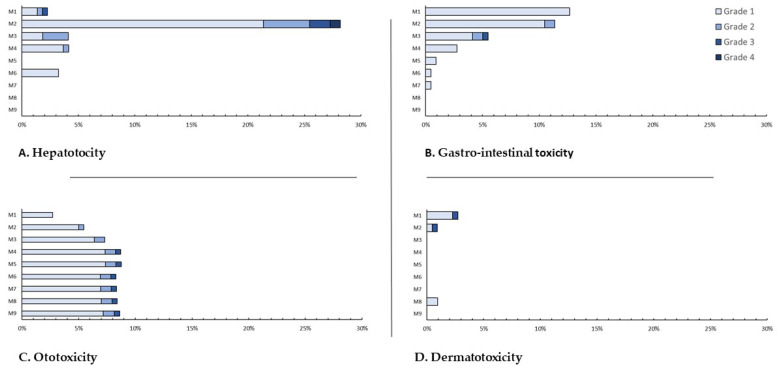
Proportion of patients with different grades of adverse event per month on treatment for rifampicin-resistant tuberculosis in Burundi. (**A**). Hepatotoxicity. (**B**). Gastro-intestinal toxicity. (**C**). Ototoxicity. (**D**). Dermatotoxicity.

**Figure 3 jcm-09-01873-f003:**
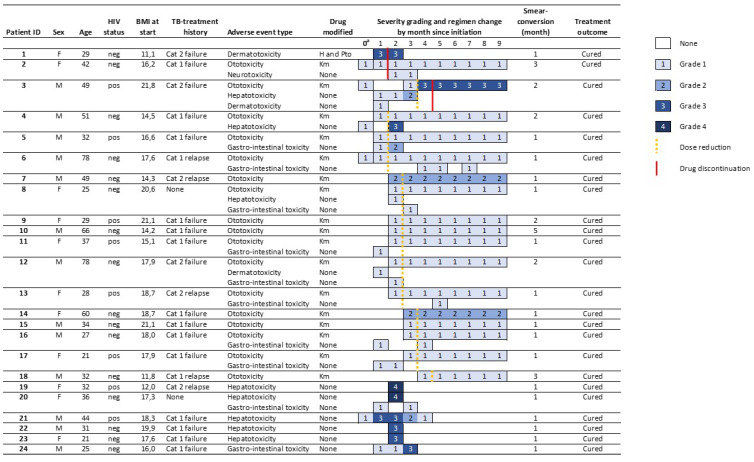
Treatment modifications and grade 3 or 4 adverse events among patients on a short regimen for rifampicin-resistant tuberculosis in Burundi. ^a^ Month 0: Abnormal test results at baseline are presented if toxicity occurred during treatment. Cat.1: six-month rifampicin-throughout treatment regimen used in new patients; Cat.2: eight-month rifampicin-throughout treatment regimen used in previously treated patients. ALT = alanine aminotransferase, AST = aspartate aminotransferase, BMI = body-mass index, F = female, H= isoniazid, Km= kanamycin, M = male, neg = negative, pos = positive, Pto = protionamide.

**Table 1 jcm-09-01873-t001:** Clinical, bacteriological and other laboratory examinations at baseline and during treatment of rifampicin-resistant tuberculosis with short regimen in Burundi.

Type of Evaluation	Month 0	Month 1	Month 2	Month 3	Month 4	Month 5	Month 6	Month 7	Month 8	Month 9	Month 15	Month 21
Clinical evaluation	x	x	x	x	x	x	x	x	x	x	x	x
Sputum smear	x	x	x	x	xx	x(x)	x(x)	x	x	xx	x	x
Sputum culture	x		x		x		x			x	x	x
Audiogram	x				x							
Chest X-ray	x											
Hemoglobin/platelet/ white blood count	x											
Serum creatinine	x		x									
Serum liver enzymes	x		x									
Electrocardiogram	xx											
HIV-test	x											
Pregnancy-test (female)	x											

x = test performed. xx = test performed twice. (x) test performed if the smear of the preceding month was positive.

**Table 2 jcm-09-01873-t002:** Characteristics of 225 patients at initiation of a short regimen for rifampicin-resistant tuberculosis in Burundi.

Characteristics	*N*	%
**Sex**		
Female	76	33.8
Male	149	66.2
**Age in years**		
<21	16	7.1
21–40	144	64.0
41–60	55	24.4
61–80	10	4.4
**HIV-status**		
Negative	179	79.6
Positive	46	20.4
Of which on ART	41	89.1
**Treatment history**		
No previous treatment	16	7.1
Cat.1 failure	136	60.4
Cat.1 relapse	30	13.3
Cat.2 failure	31	13.8
Cat.2 relapse	4	1.8
Return after loss-to-follow-up	6	2.7
Other	2	0.9
**BMI group**		
Severely underweight	59	26.2
Underweight	80	35.6
Normal	80	35.6
Overweight	3	1.3
Obese	1	0.4
Unknown	2	0.9

Cat.1: six-month rifampicin-throughout treatment regimen used in new patients. Cat.2: eight-month rifampicin-throughout treatment regimen used in previously treated patients. ART = anti-retroviral treatment, BMI = body-mass index.

**Table 3 jcm-09-01873-t003:** Baseline test results of 225 patients at initiation of a short regimen for rifampicin-resistant tuberculosis in Burundi.

Test	*N*	%
**Smear microscopy**		
Negative	2	0.9
Scanty	25	11.1
1+	85	37.8
2+	47	20.9
3+	66	29.3
**Chest X-ray**		
Normal	0	0.0
Unilateral	101	44.9
Bilateral	93	41.3
Unknown	31	13.8
**Baseline audiometry $**		
Normal	198	88.0
Grade 1 hearing loss	3	1.3
Unknown	24	10.7
**Alanine aminotransferase $**		
Normal	202	89.8
Grade 1 elevation	14	6.2
Grade 2 elevation	6	2.7
Unknown	3	1.3
**Aspartate aminotransferase $**		
Normal	192	85.3
Grade 1 elevation	24	10.7
Grade 2 elevation	2	0.9
Grade 3 elevation	3	1.3
Unknown	4	1.8
**Creatinine $**		
Normal	215	95.6
Unknown	10	4.4

$ Categories with zero patients are not shown.

**Table 4 jcm-09-01873-t004:** Outcomes of 225 patients who initiated a short regimen for rifampicin-resistant tuberculosis in Burundi.

Type of Outcome	*N*	%	Median Days (IQR) Since Initiation
**Programmatic outcomes (*n* = 225)**			
Relapse-free success	209	92.9	33 (29–61) *
Failure	1	0.4	276
Relapse	1	0.4	454
Lost-to-follow-up	3	1.3	273 (229–293)
Dead	11	4.9	81 (31–212)
**Programmatical effectiveness (*n* = 225)**			
Relapse-free success	209	92.9	33 (29–61) *
Programmatically adverse outcome ^a^	16	7.1	172 (41–266)
**Clinical effectiveness (*n* = 222)**			
Relapse-free success	209	94.1	33 (29–61) *
Clinically adverse outcome ^b^	13	5.9	114 (32–218)
**Bacteriological effectiveness (*n* = 211)**			
Relapse-free success	209	99.1	34 (29–61) *
Bacteriologically adverse outcome ^c^	2	0.9	276, 454

* Time until smear conversion. ^a^ Dead, relapse or failure, or lost-to-follow-up. ^b^ Dead, relapse or failure. ^c^ Relapse or failure. IQR = interquartile range.

**Table 5 jcm-09-01873-t005:** Adverse events among 225 patients on a short regimen for rifampicin-resistant tuberculosis in Burundi.

Type of Adverse Event	*N* Patients	%	Median Days (IQR) until Onset
**Any adverse event £**			
No event	106	47.1	
Even of any grade	119	52.9	57 (31–61)
Grade 1	92	40.9	57 (32–61)
Grade 2	18	8.0	61 (58–88)
Grade 3	7	3.1	61 (32–90)
Grade 4	2	0.9	55, 57
Grade 1/2	110	48.9	57 (32–61)
Grade 3/4	9	4.0	57 (55–62)
***N* of events per patient**			
0	106	47.1	
1	78	34.7	
2	27	12.0	
3	13	5.8	
4	1	0.4	
**Hepatotoxicity**			
No event	156	69.3	
Event of any grade	69	30.7	59 (56–62)
Grade 1	49	21.8	58 (57–62)
Grade 2	14	6.2	62 (58–92)
Grade 3	4	1.8	59 (43–62)
Grade 4	2	0.9	56 (55–57)
**Ototoxicity $**			
No event	202	89.8	
Event of any grade	23	10.2	62 (33–92)
Grade 1	20	8.9	62 (33–91)
Grade 2	2	0.9	62, 90
Grade 3	1	0.4	127
**Dermato-toxicity $**			
No event	214	95.1	
Event of any grade	11	4.9	36 (30–155)
Grade 1	10	4.4	49 (30–155)
Grade 3	1	0.4	32
**Gastro-intestinal toxicity $**			
No event	176	78.2	
Event of any grade	49	21.8	32 (29–61)
Grade 1	45	20.0	33 (29–61)
Grade 2	3	1.3	58 (25–60)
Grade 3	1	0.4	90
**Nefro-toxicity $**			
No event	224	99.6	
Event of any grade	1	0.4	88
**Neurotoxicity $**			
No event	219	97.3	
Event of any grade	6	2.7	45 (32–63)
**Osteo-articular toxicity $**			
No event	214	95.1	
Event of any grade	11	4.9	55 (29–65)

£ In case of more than one adverse event the grade shown corresponds with the highest one. $ Categories with zero patients are not shown. IQR = interquartile range.

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
