# Peer review of "Course of Adverse Events during Short Treatment Regimen in Patients with Rifampicin-Resistant Tuberculosis in Burundi"

_jcm, 2020, doi:10.3390/jcm9061873_

Round 1

Reviewer 1 Report

The manuscript present some data on Adverse events occurring over  the nine-month short-treatment regimen of TB.

The manuscript is well-written in an engaging and lively style. However, in the “introduction” section I would have wished to see more up to date references.

Author Response

Thank you for your comment. We have re-vised the introduction with reference to more relevant and/or recent publications.

Reviewer 2 Report

This study reported the treatment outcome and adverse events (AE) for short treatment regimen (STR) in patients with RR-TB.
In particular, this report describes the course of AE in detail.
Under the circumstances with limited resource, I think there are two major limitations of this study, the possibility of underestimation in the detection of AE and overestimation in the treatment success.

Major comments

1. The authors mentioned the possibility of underestimation for AE, but you should emphasize this limitation more.

2. In line 107, "Outcomes were defined per WHO guidelines [14]", and Ref 14 is WHO 2013 revision version.
In WHO 2013 version, treatment outcome for RR-TB is defined based on the results of consecutive cultures.
But, you did not report the time until culture conversion, but only report the time until smear conversion in Table 4.
And, did you use liquid media to sputum culture?
The reason for the good results seems to be based on the imprecise definition and loose test.

Minor comments

1. Even if not all patients have been tested for DST to 1st-line drugs, FQs, and second-line injectable drugs, I'd like to recommend that you present the results of the DST.

2. Page 5. There are two Table 3, same headline, different contents.

3. Line 138-139. "two (0.9%) were included without RR-confirmation"
How can these two patients be included in this study without RR-confirmation?

4. As it is a pulmonary TB study, it is recommended to report on the presence of cavities in the chest x-ray to estimate the disease severity (Table 3).

5. Table 3. The sum of the details in the item "Aspartate aminotransferase" is not 225 (191+24+2+3+4 = 224).

Author Response

This study reported the treatment outcome and adverse events (AE) for short treatment regimen (STR) in patients with RR-TB. In particular, this report describes the course of AE in detail. Under the circumstances with limited resource, I think there are two major limitations of this study, the possibility of underestimation in the detection of AE and overestimation in the treatment success.

Major comments

1.The authors mentioned the possibility of underestimation for AE, but you should emphasize this limitation more.

Response 1: Thank you for your comment. We agree that there could be an underestimation, and we propose the following changes to the discussion:
Lines 241-242: “The regimen was relatively well tolerated; severe AE were uncommonly reported, occurred early and all but one resolved quickly”
Lines 266-267: “In our cohort, overall occurrence of any reported AE (53.1%) and severe or debilitating AE (4.0%) were low”
Lines 277-278: “In Burundi, hepatotoxic events were the most commonly reported causes of having an AE and having a severe AE, yet mild forms were less commonly reported than in other studies [6,11]”.
Lines 311-313: “Part of the low reported occurrence of AE and low frequency of treatment modifications in Burundi could be explained by under detection, due to the fact that AE monitoring was less stringent compared to prospective trials.”

2. In line 107, "Outcomes were defined per WHO guidelines [14]", and Ref 14 is WHO 2013 revision version. In WHO 2013 version, treatment outcome for RR-TB is defined based on the results of consecutive cultures. But, you did not report the time until culture conversion, but only report the time until smear conversion in Table 4. And, did you use liquid media to sputum culture? The reason for the good results seems to be based on the imprecise definition and loose test.

Response 2: Thank you for this comment.
- We agree we used an outcome which is not in line with the WHO 2013 guidelines. Therefore we made a correction to the methods. These now read (line 108-110):
“We defined cured as smear converted and time until smear conversion as the number of days until the earliest of two consecutive negative smear results of at least 30 days apart, since treatment initiation. Other outcomes were defined per WHO guidelines [15]”
- Culture, when performed, was performed on solid medium, Löwenstein-Jensen. We have added this to the methods under “bacteriology”.
Line 79: “Culture was performed on Löwenstein-Jensen solid medium”
- The fact that we did not have complete culture results is indeed a limitation. We consulted with the team in Burundi on the reasons why complete culture results were unavailable.
In resource-limited settings like Burundi, it is challenging to obtain good quality culture results. Among the challenges in implementation are the availability of qualified personnel, equipment, infrastructure, and maintenance. In this particular case in Burundi, the team experienced problems with the biosafety cabinet and its maintenance. In addition, only solid medium was available for culture, which does not allow all strains to grow. The clinicians needed quick results to take clinical decisions and thus mainly relied on microscopy.
However, we believe the results are unlikely to overestimate treatment success, for the following reasons;
- Culture results were incomplete at baseline and culture was not done monthly, but all available cultures for those with relapse-free cure defined by smear results were negative at six (n=202) and nine (n=190) months.
- 176 (83.8%) of patient had a follow-up visit after one year, all were smear negative and all who had a culture result (33.5%) were culture negative.
- It is unlikely that patients with a negative smear and a good clinical presentation are suffering from treatment failure.
- The National laboratory Network has a well-functioning external quality assurance system to validate smear results
We have added a line to the discussion to emphasize the challenges related to culture in low-resource settings.
Lines 303-307: “The use of culture was limited due to problems with maintenance of the biosafety-cabinet at the national reference laboratory. Such situations are common in resource-limited settings, showing the need for adapted tools to measure treatment response [27]. However, smear-based results were likely reliable as the laboratory network has a well-established External Quality Assurance system.”

Minor comments

1.Even if not all patients have been tested for DST to 1st-line drugs, FQs, and second-line injectable drugs, I'd like to recommend that you present the results of the DST.

Response 1: Thank you, we have added a sentence in the characteristics at initiation.
Lines 140-143: “Among 60 (26.7%) patients with a baseline DST result available, DST showed RR in all (100%), resistance to H in 56/59 (94.9%), resistance to streptomycin in 14/24 (58%), resistance to E in 12/18 (66.7%), susceptibility to Km in 38/38 (100%), and susceptibility to fluoroquinolone in 37/37 (100%).”

2. Page 5. There are two Table 3, same headline, different contents.

Response 2: Thank you. This has been corrected. The title of table 4 is now; Table 4: Outcomes of 225 patients who initiated a short regimen for rifampicin-resistant tuberculosis in Burundi

3. Line 138-139. "two (0.9%) were included without RR-confirmation" How can these two patients be included in this study without RR-confirmation?

Response 3: Thank you for this comment. We consulted with the team in Burundi, who double checked the patient files. These patients had been initiated based on suspicion of RR-TB which was not confirmed by Xpert; one based on category II failure, one based on a confirmed MDR-TB- contact and no clinical improvement on TB-treatment. The phenotypic DST results for these patients delayed, and were missing in the database. These were however retrieved by the team and showed rifampicin and isoniazid resistance, and susceptibility to kanamycin and fluoroquinolones.
Lines 139-140 read: “217 (96.4%) were included based on a positive Xpert MTB/RIF, eight (3.6%) based on phenotypic DST-results only.”

4. As it is a pulmonary TB study, it is recommended to report on the presence of cavities in the chest x-ray to estimate the disease severity (Table 3).

Response 4: Thank you for this relevant comment. Indeed, it would be ideal to be able to present information on lung cavities. Unfortunately, there were no radiologists or pneumologists at the treatment centres to be able to interpret the chest X-ray, so we do not have information on lung cavities. However, we do have information on the number of lobes affected.
We have added a phrase in the characteristics at initiation.
Lines 144-146: “Among patients with an X-ray result, 59 (30.4%) showed one affected lobe only, 80 (41.2%) two or three lobes affected, and 55 (28.4%) four or more lobes affected.”

5. Table 3. The sum of the details in the item "Aspartate aminotransferase" is not 225 (191+24+2+3+4 = 224).

Response 5: This has been corrected.

Round 2

Reviewer 2 Report

Thank you for accepting my opinion.
And, I hope that the quality of this paper will be improved by accepting my opinion.